# Impact of Mountain Reservoir Construction on Groundwater Level in Downstream Loess Areas in Guanzhong Basin, China

Jia Zhang [1,2], Aidi Huo [3,4,*], Zhixin Zhao [3,4], Luying Yang [3,4], Jianbing Peng [5,*], Yuxiang Cheng [5] and Zhoufeng Wang [3,4]

1   School of Environmental Studies, China University of Geosciences, Wuhan 430074, China; flyzjsel@163.com
2   Ningxia Survey and Monitor Institute of Land and Resources, Yinchuan 750001, China
3   School of Water and Environment, Chang'an University, Xi'an 710064, China;
    zzx123guqiyongqi@163.com (Z.Z.); yangly0402@163.com (L.Y.); wangzf@chd.edu.cn (Z.W.)
4   Key Laboratory of Subsurface Hydrology and Ecological Effects in Arid Region, Ministry of Education,
    Chang'an University, Xi'an 710054, China
5   College of Geology Engineering and Geomatics, Chang'an University, Xi'an 710064, China;
    chengyx@chd.edu.cn
*   Correspondence: huoaidi@chd.edu.cn (A.H.); dicexy_2@chd.edu.cn (J.P.)

**Abstract:** An accurate understanding of the relationship between reservoir construction and the dynamic change of groundwater level in downstream areas is of great significance for rational development and utilization of water resources. At present, the research on the interaction between surface water (SW) and groundwater (GW) mainly focuses on the interaction between river and GW. There are few studies on the impact of the reservoir construction on GW level in downstream loess irrigation area. Rainfall, evaporation and climate temperature have a great impact on W level, but the impact of reservoir construction on the GW level should not be ignored in the utilization of water resources. In this paper, a GW flow model under a natural boundary was established by numerical simulation. Taking Heihe Jinpen Reservoir in Heihe River watershed as the research object, the influence of the construction of a mountain reservoir on the dynamic change of GW level in the downstream loess region is studied. By comparing the GW level under the natural state without reservoir construction and the measured GW level after the reservoir was built, the variation of the GW depth in the loess area of the lower reaches in the Heihe River watershed is obtained. The results show that simulation accuracy of the interaction between SW and GW was reasonable; after the Heihe Jinpen Reservoir construction, the mean GW level decrease was about 6.05 m in the downstream loess irrigation area in Guanzhong Basin. It provides a method for the simulation and prediction of SW–GW conversion laws. This study is also of great significance to explore the change law of the water cycle and improve the utilization rate of water resources.

**Keywords:** loess region; SWAT model; MODFLOW model; surface water; groundwater; Heihe Jinpen Reservoir

## 1. Introduction

The interaction between surface water (SW) and groundwater (GW) is not only the most common natural phenomenon, but also an important role of natural circulation. Therefore, exploring the interactive relationship between SW and GW is necessary to realize the comprehensive and standardization of water resources management [1–4]. As a special artificial lake, the relationship between a reservoir and GW is also a kind of interaction between GW and a lake. Reservoir leakage is an important reason for the interaction between a reservoir and GW [5–7]. Li et al. [8] carried out geological exploration and a sampling analysis of Beitang Reservoir in the scope of reservoir leakage and explored the GW recharge law of Beitang Reservoir leakage by the method of combining experiment with simulation. Winter [9] from the U.S. Geological Survey made a thorough study on

solving the interaction among lakes, reservoirs and GW by mathematical models. He studied the effects of different aquifer elements on their transformation based on two-dimensional and three-dimensional steady flow mathematical models, which brought great inspiration to later researchers.

The impact of reservoir scheduling and impoundment mode on GW is mainly the seepage problem caused by reservoir impoundment, the influence of GW seepage on dam stability and the influence of drainage behind a dam on soil salinization. He et al. [10] used Visual MODFLOW to simulate the changes of GW level around the reservoir before and after the completion of the riverside reservoir. It was found that the reservoir impoundment in a normal flow year reduced river infiltration and GW level, especially in July to October of flood season. The influence of reservoir impoundment on GW level was small in a dry year. Before and after the completion of the reservoir, the effect of the amount of exploitation on GW level also changed. Li et al. [11] studied the influence of Mushroom Lake Reservoir and Daquangou Reservoir leakage on GW level in Manas River Basin by the method of isotope technology and numerical simulation. The result showed that the greatest influence range of reservoir leakage was 9000 m downstream of the reservoir. The GW recharge to the reservoir was affected by both the reservoir level and the GW level, but the model boundary conditions she established were artificial around the reservoir. The model needs to be improved to be more consistent with the natural boundary conditions.

A reservoir is a SW body controlled by humans; the interaction between a reservoir and GW is a complex process. In addition to the simple GW movement model, scholars also coupled the basin-scale hydrological model with the GW model [12–14]. As the input value of GW recharge, the output of the hydrological model not only improves the accuracy of the model, but also reduces the error caused by the selection of recharge value.

Sepideh [15], Sophocleus [16], Liu et al. [17], Zhang et al. [18] and other researchers couple the The Soil & Water Assessment Tool (SWAT) model and MODFLOW model to improve the recharge module of the GW model. Compared with a single model, the accuracy and precision of prediction of the coupled model have been greatly improved. However, each model may have different calculation units and discretization methods, so the scale matching problem restricts the development of coupled models. The rapid development of Geographic Information System (GIS) not only plays an important role in the prediction and evaluation of water resources, but also provides convenient conditions for GW numerical simulation [1,19–21].

It can be seen from the existing literature that the interaction between SW and GW has always been the focus of hydrologists [22–24]. An accurate understanding of the relationship between SW and GW plays an important role in the rational development and utilization of water resources [24–26]. However, most studies reflect the interaction between SW and GW through the interaction between river and GW. The research on the interaction between reservoirs and GW mainly focuses on the safety problems caused by reservoir leakage and the impact on soil salinization. There are few studies on the impact of mountain reservoir storage on GW level in downstream loess irrigation area.

However, due to the uneven temporal and spatial distribution of water resources in the loess area of Guanzhong Basin, reservoir storage is a common measure to alleviate the water pressure caused by it. The construction of the reservoir has changed the water utilization structure of the basin, resulting in changes in the recharge and discharge conditions of GW. The impact of reservoir storage on GW level should not be ignored. Since the birth of numerical simulation technology, researchers in various countries have improved it, and the effect and accuracy have been greatly improved and recognized. Therefore, this paper establishes a GW flow model under a natural boundary by means of numerical simulation. Taking Heihe Jinpen Reservoir in Heihe watershed as an example, it is scientific and reasonable to study the effect of reservoir regulation and storage process in mountainous areas on the dynamic change of GW levels in the downstream loess region. The research is helpful to explore the influence of reservoir construction on the water circulation path change and improve the utilization rate of water resources in the Guanzhong Basin.

## 2. Study Area

This study was conducted in the Heihe River Basin (E 107°43′~108°24′, 33°42′~34°13′ N), in the upstream of the Jinpan Reservoir, covering an area of 2258 km². The Heihe River originates from the valley near Mafu Town, Zhouzhi County, Shaanxi Province and finally flows into Weihe River in Shima village. The upper reaches of Heihe River pass through Qinling Mountains, forming Qinling an alluvial-proluvial fan in the middle and finally flows into Weihe River in the lower reaches. Its terrain tilts from southwest to northeast. The upper reaches of the Heihe River pass through the Qinling Mountains, forming Qinling alluvial fans in the middle and finally flowing into the Wei River in the downstream area [27]. Its elevation is high in the southwest and low in the northeast [27,28]. Through numerical simulation of the interaction between GW and GW, this paper focuses on the change of GW level after the construction of the reservoir in the Heihe River downstream loess region (Figure 1) and analyzes the effect of the reservoir on GW level in the study area.

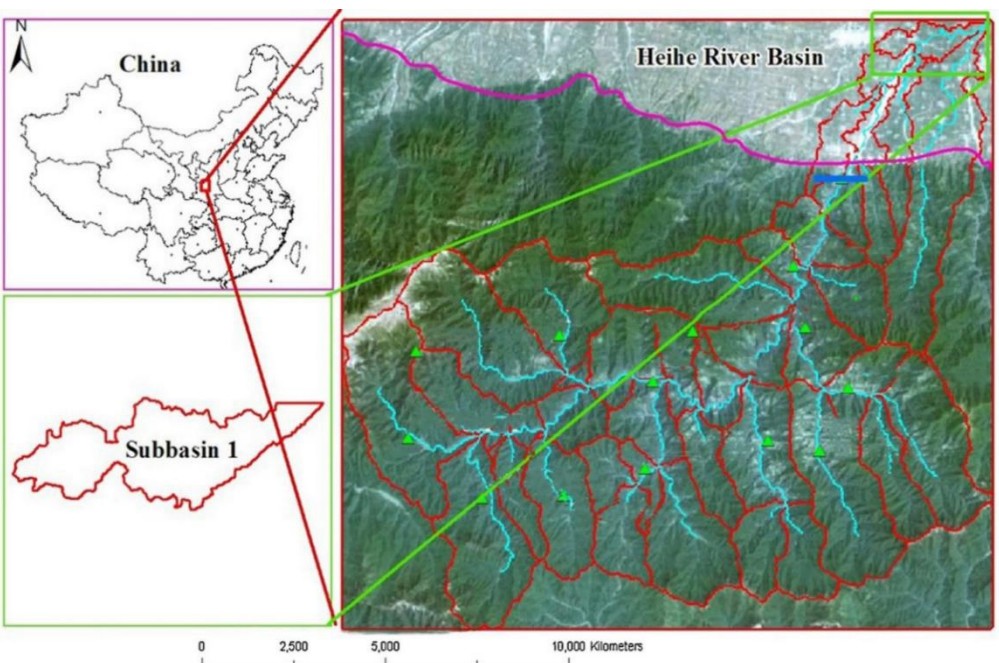

**Figure 1.** Location map of study areas in Guanzhong Basin [29].

Heihe Jinpen Reservoir is located about 1.5 km above Heihe River Yukou, 86 km away from Xi'an city. It is a large-scale water hydraulic project with a comprehensive utilization of urban water supply, power generation, flood control and agricultural irrigation, etc. It consists of a barrage, a spillway tunnel, a water diversion tunnel, a power station behind the dam and an ancient watercourse seepage control project. Heihe Jinpan Reservoir, completed in 2008, with a maximum depth of 83 m, is a reservoir area of 4.55 square kilometers and a total storage capacity of 200 million cubic meters. The reservoir is designed according to the once-in-a-century flood standard (Q = 3600 m³/s) and checked by the once-in-two-millennia flood standard (Q = 6400 m³/s). The normal high-water level is 594.0 m, and the flood limit water level is 593.0 m. The designed and checked flood levels are 594.34 m and 597.18 m, respectively.

There are clear boundaries between mountains and rivers in the Heihe River Basin. Mountain area and valleys are above Heiyukou with a catchment area of 1481 km², accounting for 65% of the whole basin. The total length of Heihe River is 91.2 km. The average slope of watercourse is 14.7‰. The river system is dendritic. The average width of Heihe Basin is 16.2 km. Tributaries are more concentrated on the right bank, and the catchment area of the right bank is three times as that of the left bank. After Hei River outflows from Yukou, it is nearly a flat area with a riverbed gradient about 1/240~1/1280. There are many

mountains with steep slopes, dense tributaries and exposed rock in the Heihe River Basin. The mountains are covered with about 25~50 cm of yellow-brown sandy soil and black humus, which are from the weathered rock. Watershed vegetation to prolong the service life of the reservoir and the forest coverage rate is about 26.5% above the Yukou.

The temporal and spatial variation of annual precipitation in Heihe River Basin is quite different, with an average annual precipitation of 810 mm. The heavy precipitation from July to October accounts for more than 60% of the annual precipitation [29]. The annual precipitation of remote mountainous areas in the north of Qinling Mountains is less than 800 mm, and that of other mountainous areas is about 1000 mm. The Heihe Jinpen Reservoir, built in 2008, not only provides drinking water for 8 million residents in Xi'an, but also provides irrigation water for 24,700 km$^2$ farmland areas. Due to the abundant water supply and the increasing population with the development of the city, Xi'an needs more and more water. The elevation of the basin is 260~3754 m [24,30,31].

## 3. Methods

We obtained the variation of the GW level after Heihe Jinpen Reservoir construction in the Heihe River downstream loess region through comparing the underground water level without reservoir construction in the natural conditions with the measured GW level after reservoir construction in order to analyze the influence of Heihe Jinpen Reservoir construction on the GW level. The GW level in a natural state can be realized by coupling the SWAT and MODFLOW models after which it is verified and corrected (SWAT was developed by Dr. Jeff Arnold, Agricultural Research Center, USDA in 1994. MODFLOW is a three-dimensional groundwater simulation model developed by the United States Geological Survey.). The measured water level was obtained through field survey. The time of numerical simulation is defined according to that of the field investigation. A framework of numerical simulation is shown in Figure 2.

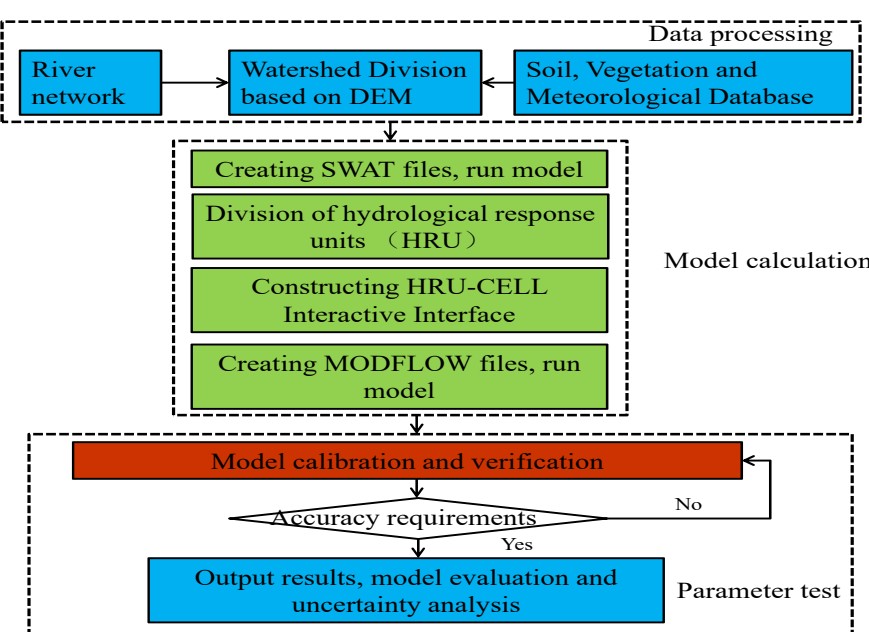

**Figure 2.** Framework of numerical simulation for GW depth without reservoir construction.

### 3.1. Numerical Simulation Method of GW Level under Natural Conditions

SWAT is a physical-based continuous-time model with high computational efficiency and can simulate a long time based on a basin scale. The major input data include weather, hydrology, soil temperature and properties, plant growth, nutrients, pesticides, bacteria and pathogens and land management. In SWAT, a larger basin is divided into several

sub-basins. Then, it is further subdivided into hydrological response units (HRUs), which have the same land use, management and soil composition [32,33].

ModFlow [34] is a three-dimensional GW simulation system with a physical basis. It combines Darcy's law with mass balance of underground flow. Generally, all the common boundary conditions can be considered in the model, including confined, unconfined, leaky, delayed yield and variably confined/unconfined conditions. Both steady state and transient conditions can be simulated.

The centralized characteristic of the SWAT model makes its performance of handling GW flow limited, and MODFLOW is limited in handling distributed GW recharge. Through the combination of the two advantages, it can better reflect the temporal and spatial characteristics of the basin. The specific operation is to use the GW recharge based on HRU as the input data of MODFLOW to calculate and exchange the GW flow between aquifer and water system to SWAT [35].

There are many factors affecting GW simulation. GW recharge and evaporation are the main factors directly affecting GW simulation, but it is usually difficult to estimate accurately. In order to simulate the change process of GW in detail, the GW simulation module uses the module in MODFLOW. Since the distributed SW recharge in the MODFLOW is not as detailed as that in the SWAT, the linkage may consist of modifying both the SWAT and MODFLOW original programs. The following will introduce how GW partial conversion realizes the transition from the SWAT model to MODFLOW models [1,36].

Hydrological response unit (HRU) is the basic calculation unit of simulation in a distributed SWAT model, which is the comprehensive impact of sub watershed soil cover and land use. In the SWAT preprocessing menu, the land use and soil vector map has its own spatial location and can be exported as a SHP file. Then, it is calculated and analyzed through the spatial superposition tool of ArcGIS 10.3 software (ArcGIS is a series of client software, server software and online geographic information system services developed and maintained by the American Institute of Environmental Systems Research Institute), and SWAT generates a complete HRU file. This file contains information on the combination of soil and land use attributes, which provides a reference for establishing the spatial location of watershed HRU. The soil and land use map is exported to a shape file, and its spatial location number can be obtained by determining the cell through Digital Elevation Model (DEM), data sources, which are shown in Table 1. Then, the HRU numbers in the sub slot in the SWAT output complete HRU file are read, and these numbers are assigned to the location of the corresponding unit through MODFLOW. The number in the grid corresponds to the HRU number of the SWAT model; the number 0 indicates the area outside the boundary of the MODFLOW model. Through the above position correspondence, the coupling between SWAT and MODFLOW can be reasonably realized [1]. As the GW recharge rate of HRU in the sub watershed in SWAT lacks spatial information, it is necessary to obtain the spatial distribution map of HRU through the ArcGIS spatial superposition function, then allocate the recharge value of HRU in SWAT to each unit, and then allocate the HRU recharge value to the cell of MODFLOW. The HRU number is used as the partition number in the "basic" package of MODFLOW [35]. Figure 3 shows the schematic diagram of GW recharge computation in SWAT–MODFLOW [1].

**Table 1.** The simulated flow results in the Chenhe station.

| Station Name | Time | $R_{vol}$ | $r$ | *Ens* |
|---|---|---|---|---|
| Chenhe hydrological station | 1 January 2005–31 December 2008 (Calibration period) | 0.020 | 0.89 | 0.86 |
| | 1 January 2009–3 April 2013 (Validation period) | 0.019 | 0.90 | 0.82 |

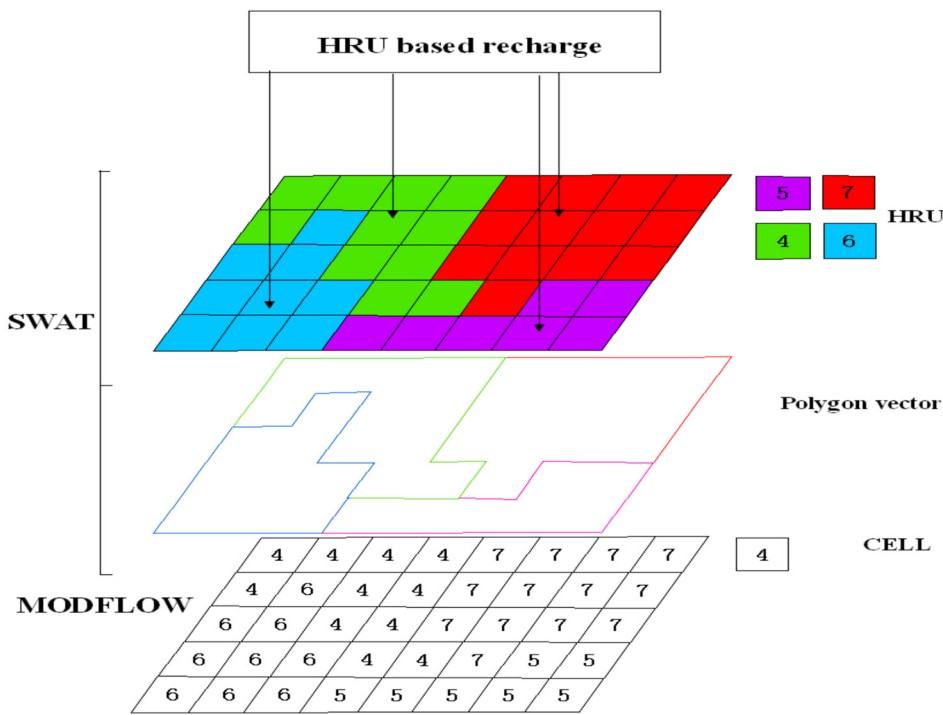

**Figure 3.** Schematic diagram of recharge computation in SWAT–MODFLOW [1].

### 3.2. Field Investigation

In order to collect accurate data that can be used to support the model, the investigation team not only consulted the relevant information of villages and towns in Heihe River Basin, but also asked the relevant information of local government by telephone, so as to select the appropriate exploration route on the map. On 18 October 2014, the investigation team investigated 38 groups of GW level data in sub watershed 1 of Heihe River Basin. Measuring rope and a general electric energy meter (ABSD-802-GPRS produced by Shandong Oberside Automation Technology Co., Ltd. in Shandong Province, China was selected) were the main measuring tools. The operation process was as follows: (1) continuously put the measuring rope into the well until the pointer of the multimeter deflects; (2) read the scale on the measuring rope to obtain the current water level.

### 3.3. Model Calibration and Verification

In this paper, the SW in the SWAT model was calibrated and verified. We used the water balance coefficient *(rVol)*, efficiency coefficient (*Ens*) and correlation coefficient (*r*) as objective functions to evaluate the model [37,38] (Table 2).

**Table 2.** The hydrogeological parameters after MODFLOW model calibration.

| Location | | Horiz Conductivity/m/d | Vertical Conductivity/m/d | $S_y$ Specific Yield | $S_s$ Storativity |
|---|---|---|---|---|---|
| River area | The first aquifer | 8.3 | 0.83 | 0.2 | $1 \times 10^{-4}$ |
| | The second aquifer | 0.01 | 0.001 | 0.08 | $1 \times 10^{-4}$ |
| Irrigated area | The first aquifer | 5 | 0.5 | 0.04 | $1 \times 10^{-4}$ |
| | The second aquifer | 0.01 | 0.001 | 0.08 | $1 \times 10^{-4}$ |

Note: the first aquifer is a shallow aquifer group with microconfined properties whose bottom buried depth is less than 50 m in the distribution area of submersible and confined water. The second aquifer is a confined water area, and the bottom boundary of the aquifer group is 80~120 m deep, with bedrock as the bottom boundary in some areas [39].

## 4. Results and Discussion

### 4.1. GW Depth Change

The typical measured GW depth of GW in 20 wells, while there is no GW at the bottom of the other wells, are shown in Table 3. From the measured data and the feedback of local residents, we can briefly and preliminarily conclude as follows: (1) the water depth of the wells is approximately 10 m below, and the water level of the wells closer to the Heihe Reservoir is deeper. Several wells around the Heihe Reservoir have dried up, while the situation in distant villages and towns is slightly better due to the replenishing of the Weihe River and rainfall. (2) The construction of reservoirs has a great impact on the surrounding villages and towns. In the past, villagers generally used well water to meet their living needs, but now they basically or completely rely on water towers built in villages and towns to obtain water for reference and living. (3) The quality of the well water decreased greatly compared with that after the construction of the reservoir. Some villagers reported that before the reservoir was built, the well water was clear, drinkable and tasted sweet.

**Table 3.** Typical GW Depths of field investigation wells.

| Number | Altitude/m | GW Deph/m | Survey Location |
|---|---|---|---|
| 1 | 419 | 8.76 | Xinjiazhai Village |
| 2 | 396 | 11.755 | Wuhe Village, Xinjiazhai |
| 3 | 423 | 6.28 | Four Groups of Weiyou Village in Xinjiazhai |
| 4 | 413 | 10.58 | 5 Groups of Weiyou Village in Xinjiazhai |
| 5 | 377 | 8.75 | Group 7, Weixing Village, Furen Township |
| 6 | 415 | 7.18 | Group 13, Weifeng Village, Furen Township |
| 7 | 436 | 9.03 | Four Groups of Hengzhou Village, Xinjiazhai |
| 8 | 456 | 8.79 | Group 2, Hengzhou Village, Xinjiazhai |
| 9 | 428 | 7.4 | Group 2, Wangtang Village, Sizhu Township |
| 10 | 429 | 7.76 | Group 3, Nansi Zhucun, Sizhu Township |
| 11 | 432 | 8.13 | Six Groups of Beisizhu Village, Sizhu Township |
| 12 | 405 | 5.68 | Group 5, Beisizhu Village, Sizhu Township |
| 13 | 428 | 4.73 | Group 3, Beisizhu Village, Sizhu Township |
| 14 | 418 | 4.00 | Longquan Temple |
| 15 | 427 | 7.92 | Group 4, Nanqi Village, Sizhu Township |
| 16 | 449 | 12.26 | Group 8, Shangcun, Louguan Town |
| 17 | 320 | 19.03 | Louguan Town Sanjiazhuang 8 Team |
| 18 | 437 | 24.57 | Group 2, Dongjiayuan Village, Louguan Town |
| 19 | 428 | 3.5 | Heihe bridge head |
| 20 | 430 | 4.3 | Ma Zhao Xiang |

Note: There is GW at the bottom of all the wells.

### 4.2. Modeling Results

Figure 4 shows the distribution of Heihe River recharge and potential evapotranspiration on 10 October 2014. It can be seen from Figure 4a that the recharge on the north bank of Heihe River is higher than that on the south bank, and the recharge is different between different sub-basins, with obvious differences. Among them, the sub-basins with the largest GW recharge value are the 15th and 22nd, followed by the plain irrigation area. It can be seen from Figure 4b that the largest potential evapotranspiration is concentrated in the 6th, 7th and 42nd basins, followed by the downstream plain area and the south bank of Heihe River (Heihe Jinpen Reservoir located in the seventh basin). In mountainous areas, potential evapotranspiration on the south bank of Heihe River is lower than that on the north bank, which may be related to GW depth and lithology of the unsaturated zone. As potential evaporation is affected and restricted by many factors, such as temperature, precipitation, water pressure, lithology, vegetation and crops, its change and movement process are very complex. For this purpose, several observation wells are set in the basin (Figure 5).

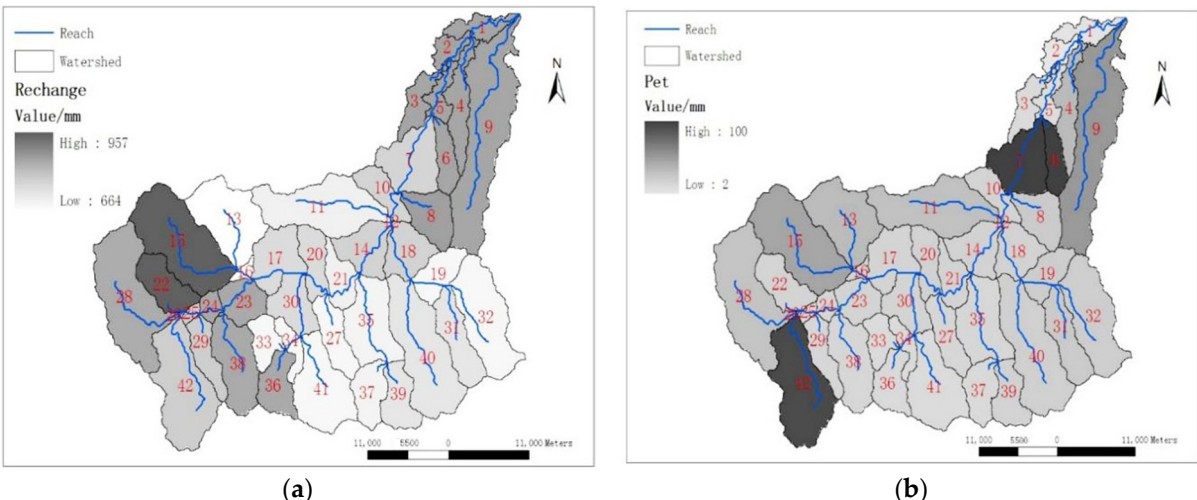

**Figure 4.** (**a**) Recharge and (**b**) potential evapotranspiration on 10 October 2014.

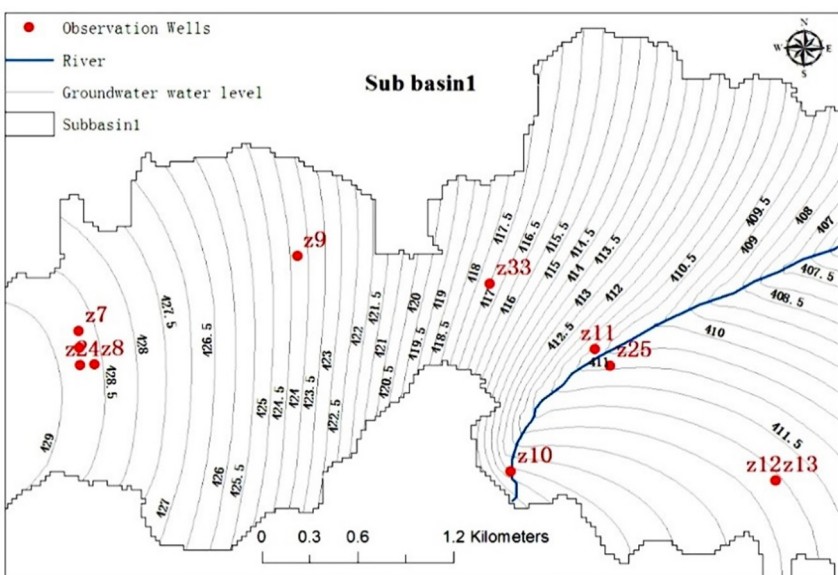

**Figure 5.** The spatial position of the measured GW sites.

Figure 5 is the distribution diagram of simulated GW level. The results show that the change of GW level in the first basin is bounded by the east and west of Heihe River; that is, in the west of Heihe River, the GW level increases gradually from southeast to northwest; in the east of Heihe River, the GW level gradually decreases from south to north. In GIS, the simulated depth of GW (DEM minus GW level) is obtained by using the spatial analysis tool, and then the measured depth of GW is obtained by measuring the depth of observation logging in the field. The simulated and measured values are in good agreement. It was also found that the depth of GW near Heihe River is very shallow and gradually increases along the northwest and southeast.

### 4.3. Contrastive Analysis of Simulated Value and Measured Value

It can be seen from Figure 6 and Table 4 that the correlation coefficient between the simulated value and the measured value is 0.88, indicating that the error in the modeling process is small, and the groundwater closure mode of the two is basically the same, which shows that the parameters and conditions used in the modeling process are generally reasonable and correct. The model can be used to simulate the irrigation water cycle.

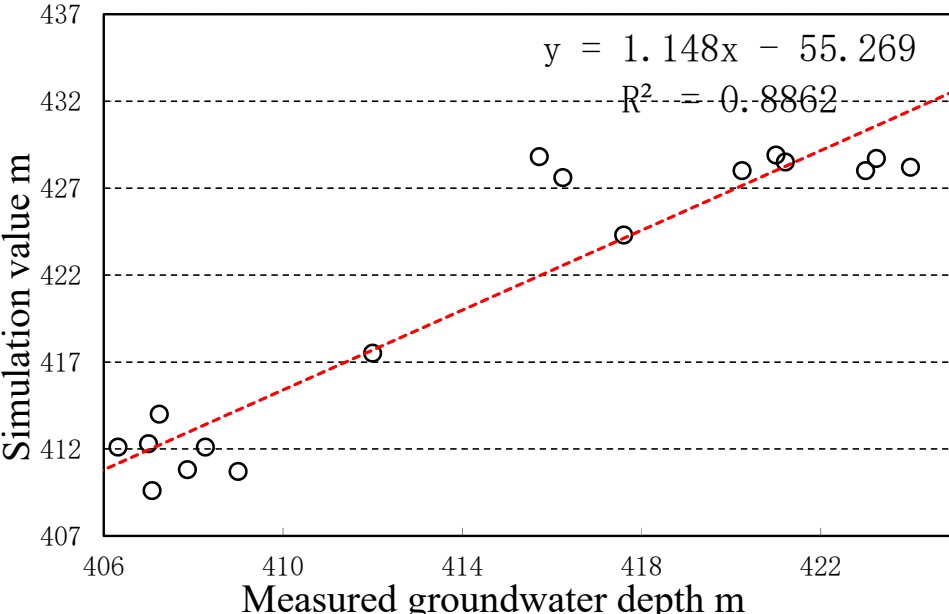

**Figure 6.** Scatter diagram of simulated and measured GW level in sub-basin 1.

**Table 4.** Calculated underground water depth and observation value error indicators (MODFLOW model).

| Indicators | Average Value/m | VAR/m | STDEV | R | Average Error Value/m |
|---|---|---|---|---|---|
| Simulated values | 421 | 67.2 | 8.2 | 0.88 | 6.05 |
| Observed values | 415 | 45.2 | 6.7 | | |

## 5. Discussion

According to Figure 6 and Table 4, the measured value of groundwater depth is 6.05 m lower than the simulated value. The main reason is that the impact of Heihe Jinpen Reservoir water diversion on Xi'an city was not considered in the simulation process (Figure 1, blue line position and Figure 7). According to the field survey, the water depth of the observation well increases with the increase of the distance from the reservoir in the lower reaches of Heihe River. For example, several wells close to the reservoir in the lower reaches of Heihe River have completely dried up; wells in several villages far away from the reservoir did not run dry because of rainfall and the Weihe River recharge. At the same time, the construction of the reservoir will also have a certain impact on the surrounding residents. For example, before the construction of the reservoir, the villagers' domestic water depended on well water, which tastes very sweet. After the completion of the reservoir, the villagers can only live on the water tower, and the water is turbid most of the time.

As is well known, the decline in GW levels will not only cause the pumping wells of agricultural motors to be suspended, resulting in the abandonment of the pumping wells, but also greatly reduce the soil moisture in the tillage layer, bringing a series of ecological and environmental problems.

It can be seen from the results of the GW numerical simulation of the No. 1 sub-basin in 2014 that the simulation trend is basically consistent with the measured results. However, the simulation trend may also change due to differences in the study area. The coupled computational model in this study can more accurately simulate and predict the changes of GW resources, providing scientific methods for regional water resource planning and scientific management, which is of practical significance for achieving sustainable use of regional water resources.

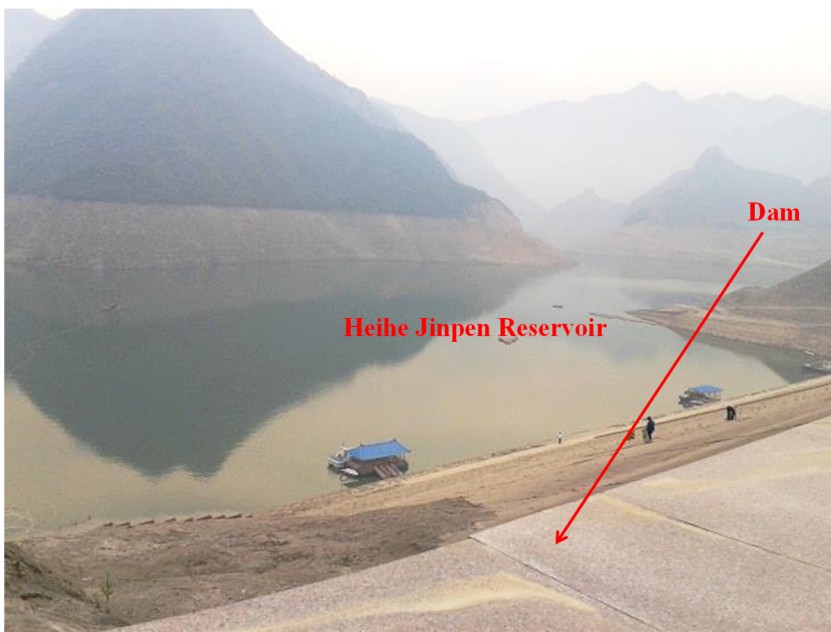

**Figure 7.** Heihe Jinpen Reservoir and Dam.

However, in the process of popularization and application, the researchers should be aware of the following problems: the dielectric constants of different soil types correspond to soil moisture differently, and remeasurement is required to improve moisture content accuracy (note: the research object in this paper is the loess irrigation area in the lower reaches of Heihe River). Adding the motion law of the aeration zone into the coupling model can show the physical mechanism in more detail, which requires our perfect processing process. After the reservoir construction in mountainous areas, the following problems should be paid attention to:

(1) Water quality, GW level and water quality monitoring websites near reservoirs should be set up in time to monitor the water level, quality of reservoirs and nearby GW. The changes of GW level, water quality in reservoir and its vicinity should be grasped in time to ensure good water quality of reservoir.

(2) Impervious work in the reservoir area should be done strictly. Strict anti-seepage treatment should be carried out on both sides of large fissures; agricultural irrigation, rural drinking water and other observation wells and exploration wells should be blocked in the reservoir submerged and relocated area to prevent reservoir water from being injected into wells, which will have a connection with aquifers, increase the leakage of reservoirs and pollute the aquifers.

(3) Establish an emergency response mechanism for risk accidents. To formulate emergency treatment measures, such as closure and closure of sewage leakage points in upstream reservoirs under sudden accidents, centralize sewage collection and treatment and prevent sewage from entering reservoirs and polluting GW.

(4) Strengthen the construction of protective forests around reservoirs and corresponding management measures, increase the green area of upstream areas of reservoirs so as to conserve water sources and purify water quality.

## 6. Conclusions

In the loess area of the downstream Heihe River, the results of the integrated simulation reveal that: (1) the interaction model of SW and GW in the irrigation zone was established; (2) the mismatch problems were solved by corresponding the HRUs and the cells. According to the principle of the SWAT model, the HRUs and the cells were mapped into a finite difference grid, and then the calculation results of the SWAT model were imported to obtain the coupling calculation model; (3) the GW level simulation results in this study area are

basically consistent with the real results, indicating that the simulation accuracy is reliable; (4) it provides a reference method for the simulation and prediction of SW–GW conversion law; (5) the simulation analysis showed that the average value of GW level in the loess area in the lower reaches of Guanzhong Basin has declined by 6.05 m since the construction of Heihe Jinpen Reservoir. Reservoir construction in mountainous areas reduces the recharge of GW in the downstream, lowers the GW depth near the reservoir and downstream areas and increases the difficulty of agricultural irrigation exploitation. Reservoir construction in mountainous areas has a greater impact on the normal exploitation of downstream water sources.

**Author Contributions:** Conceptualization, J.Z. and A.H.; methodology, Z.Z.; software, L.Y.; validation, Z.Z., L.Y. and Y.C.; formal analysis, Z.W.; investigation, Z.W.; resources, Y.C. and J.P.; data curation, J.Z.; writing—original draft preparation, J.Z.; writing—review and editing, Z.Z.; visualization, Y.C.; supervision, Z.W.; project administration, A.H. and J.P.; funding acquisition, A.H. All authors have read and agreed to the published version of the manuscript.

**Funding:** This work was supported by the National Natural Science Foundation of China (Grants Nos. 41877232 and 41790444); Natural Science Foundation of Ningxia, China (Grants Nos: 2021AAC03429, 2021AAC03426); the sixth Lift Engineering of young scientific and technological talents of Ningxia, China (Grants No: NXKJTJGC2021074); the Fundamental Research Funds for the Central Universities (Grant No. 300102278113); the Key Laboratory Open Project Fund of State Key Laboratory of Loess and Quaternary Geology, Institute of Earth Environment, CAS (Grant No. SKLLQG1909); the Public Welfare project of Ningbo City (Grant No.202002N3139); and the Key R & D plan of Shaanxi Province (Grant No. 2020SF-424).

**Institutional Review Board Statement:** Not applicable.

**Informed Consent Statement:** Not applicable.

**Data Availability Statement:** Not applicable.

**Conflicts of Interest:** The authors declare that they have no known competing financial interest or personal relationships that could have appeared to influence the work reported in this paper.

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
