# Peer review of "Impact of Mountain Reservoir Construction on Groundwater Level in Downstream Loess Areas in Guanzhong Basin, China"

_water, doi:10.3390/w14091470_

Round 1

Reviewer 1 Report

The Authors present a potentially interesting research study dedicated to investigating the downstream effects of dam. The main substantive focus of the paper is directed toward GIS and numerical modeling of the impact of an artificial reservoir (the Jinpen Reservoir) on the groundwater level. According to the introduction of the paper, the ecosystem security in the Guanzhong Basin as well as the impact of the reservoir on the ecological environment was also to be an important part of this work. In my opinion, not all of the stated goals of this study have been achieved. Certainly, the environmental impact of such reservoirs has not been analyzed. Missing from the introductory chapter is an overview of work in this area (e.g. temperature modification in impounded rivers, downstream effects of dams induce sediment transport and river channel sedimentation and erosion balance; the changes in the longitudinal connectivity of river reaches, and many other, often related negative impacts). Environmental issues are only discussed perfunctorily in the discussion of results (Section 4.3) and are more likely to be recommendations for water quality monitoring and rules for use of the reservoir environment. However, the issues discussed in this chapter are not directly derived from the results. In summary, it is difficult to understand the clear purpose of this work. Furthermore, serious drawbacks in the descriptions of the study site and methods, do not allow for publishing it in Water in the present form. More detailed comments on these sections are provided below.

Study area

Line 30: The words in the title of the manuscript should not be duplicated in keywords.

Lines 108-111: What is the source of this data (geomorphology)? Maybe it is worth providing references to this information?

Line 111: Please replace the word topography – with relief or elevation.

Line 116: Figure 1 is illegible; Please change the scale bar to readable, correct legend (no abbr., every feature should be named). The figure depicting the study area is extremely important for interpreting the results. Too much different content here, while the names referred to by the authors in the text are not presented, e.g. the Chenhe station, Yukou, ...

It is also worth standardizing the name of the reservoir; in the manuscript the names: Jinpen Reservoir, Heihe reservoir, Heihe Jinpen Reservoir, are used interchangeably. This creates unnecessary confusion and problems in understanding the context.

Lines 122-126: Key parameters have been omitted, it is difficult to understand the scale of the problem, e.g. original gross storage, surface area, mean depth, and extremely important information - when the construction/first filling of the reservoir was completed.

Lines 136-137: Watershed vegetation is good and … - What does it properly mean?

Lines 138-141: Please add references to this information? What time period does the precipitation data cover?

Lines 142-144: In the end, I found the date of construction of this reservoir. Nevertheless, in relation to this whole sentence, it is worth adding information on whether the demand for water in the agricultural sector and for the needs of Xi’an City residents has tapered off significantly. I think this will be a good background for the final discussion.

Methods

Lines 151-152; 187: Please include both the manufacturer's name and location (including city, state, and country) for specialized software (SWAT, MODFLOW, and ArcGIS).

Rather, Section 3.1 is a description of software principles and computational methods. What is missing here is information on the source data for the DEM/DTM as well as the databases used for soils, vegetation, and meteorological features. Please consider indicating how actual this dataset is, and above all, how accurate it is. Especially the DEM (or DTM) should have a precisely described vertical and horizontal accuracy. The quality of the material may be the basis for the discussion of the results.

Lines 212-214: I realize that measuring the position of the water table in a well is a relatively simple operation, however, please explain what type of device (general electric energy meter/ multimeter) was used to do this.

Lines 218-219: Shouldn't this table be in the results? These results are worth discussing.

Altitude/m > Altitude [m a.s.l.]

Some small points include:

The manuscript contains a lot of typographical errors, which makes it difficult to read. The quality of the presentation also needs improvement. The reference list and style of citations need to be adapted to the MDPI Water requirements.

Author Response

As the comments in the reply involve figures and tables, we have uploaded Word in the attachment.

Reviewer 2 Report

attached, pleas find  the word file with corrections an suggestions 

Author Response

Reply:

Thank you for your great commands. The following is our explanation for the revision of the manuscript:

  1. We have changed the “Marios” into “Sophocleus” at line 73.
  2. We have removed the redundant different and modify the sentence to “each model may have different calculation units and discretization methods”.
  3. Yes, we superscripted the 2 at line 107.
  4. We have changed the “et al” into “etc.” at line 124.
  5. Reply:Thank you for your great commands.

The reservoir is designed according to the once-in-a-century flood standard (Q=3,600 m3/s) and checked by the once-in-two-millenniums flood standard (Q=6,400 m3/s). According to Zhao et al. (2013), the first aquifer is a shallow aquifer group with micro-confined properties, whose bottom buried depth is less than 50 m in the distribution area of submersible and submersible and confined water. The second aquifer is confined water area, and the bottom boundary of the aquifer group is 80 ~120 m deep, with bedrock as the bottom boundary in some areas. We have commented it at the bottom of Table 2.

  1. We have changed the “2.47 x 104” into “24700” at line 149.
  2. We have changed the “Swat” into “SWAT” in the text.
  3. Thank you for your great commands. We have changed “manuscript” into “file” at line 198.
  4. Thank you for your great commands. We added the literature and checked the references for the full text.
  5. According to Zhao et al. (2013), the first aquifer is a shallow aquifer group with micro-confined properties, whose bottom buried depth is less than 50 m in the distribution area of submersible and submersible and confined water. The second aquifer is confined water area, and the bottom boundary of the aquifer group is 80 ~120 m deep, with bedrock as the bottom boundary in some areas. We have commented it at the bottom of Table 2.

Zhao W, Lin J, Wang SF, et al. (2013). Influence of Human Activities on Groundwater Environment Based on Coefficient Variation Method. Environmental Science, 34, 1277-83. doi:10.13227/j.hjkx.2013.04.040.

  1. yes we changed the “pet” into “potential evapotrnaspiration” in the text.
  2. We have deleted and replaced " It can be seen ", and the revised sentence is “The simulated and measured values are in good agreement. It is also found that the depth of GW near Heihe River is very shallow and gradually increases along the northwest and southeast.”
  3. We elaborate on the sentence, changing "are better" to " did not run dry because of rainfall and the Weihe river recharge" at line 284.
  4. Yes, we deleted “in Shaanxi Province, China.” at line 305.
  5. we deleted the " underground" at line 317 and 318.
  6. Reply:Thank you for your great commands. Our opinion is consistent with your suggestion, we changed the “groundwater level” into the “groundwater depth” at line 354.
  7. We have modified the references as follows:

Kim NW, Chung IM, Won YS, Arnold, J.G. (2008) Development and application of the integrated SWAT-MODFLOW model. Journal of Hydrology 356 (1-2):1-16.

Round 2

Reviewer 1 Report

The revised manuscript was improved from the previous version. However, there are still some serious flaws in this paper. The restructuring manuscript does not address the important issues raised in my previous review. The purpose of the work is still unclear. The authors only addressed the specific comments, and completely missed the aspects raised at the beginning of my review. In line with my previous review, one of the objectives included in the introduction is the impact of artificial reservoir construction on the environment and ecological environment in the study area (cf: line 31, line 90, and especially 112-113). This article deals only with water resources and does not resolve issues related to ecosystem protection or the ecological environment in the study area. This issue needs to be explicitly clarified by supplementing the introduction and discussion of the results, or by rewording the points raised above. Since biotic aspects were not the subject of the study done by the authors, I think the second approach will be more beneficial.

Author Response

Reply:
Thank you for your great commands.
We have rewritten the relevant content of the article according to your great commands, as follows:
In line 30-31, we have rewritten the sentence to “This study is also of great significance to explore the change law of water cycle and improve the utilization rate of water resources.”
In line 99-101, we have rewritten the sentence to “The research is helpful to explore the influence of reservoir construction on the water circulation path change and improve the utilization rate of water resources in the Guanzhong Basin.”
In line 114, we have deleted “and ecological environment”.

This manuscript is a resubmission of an earlier submission. The following is a list of the peer review reports and author responses from that submission.